# Online Safety for Children and Youth under the 4Cs Framework—A Focus on Digital Policies in Australia, Canada, and the UK

**DOI:** 10.3390/children10081415

**Published:** 2023-08-19

**Authors:** Yujin Jang, Bomin Ko

**Affiliations:** 1Department of Early Childhood Education, College of Social Science, Gachon University of Korea, Seongnam 13120, Republic of Korea; yjjang@gachon.ac.kr; 2Division of International Trade, College of Commerce and Public Affairs, Incheon National University, Incheon 22012, Republic of Korea

**Keywords:** online safety, case study, comparative study, children, youth

## Abstract

This study analyzes the previous literature on the online safety of children and youth under “the 4Cs risk framework” concerning contact, content, conduct, and contract risks. It then conducts a comparative study of Australia, Canada, and the UK, comparing their institutions, governance, and government-led programs. Relevant research in Childhood Education Studies is insufficient both in quantity and quality. To minimize the four major online risks for children and youth in cyberspace, it is necessary to maintain a regulatory approach to the online exposure of children under the age of 13. Moreover, the global society should respond together to these online risks with “multi-level” policymaking under a “multi-stakeholder approach”. At the international level, multilateral discussion within the OECD and under UN subsidiaries should continue to lead international cooperation. At the domestic level, a special agency in charge of online safety for children and youth should be established in each country, encompassing all relevant stakeholders, including educators and digital firms. At the school and family levels, both parents and teachers need to work together in facilitating digital literacy education, providing proper guidelines for the online activities of children and youth, and helping them to become more satisfied and productive users in the digital era.

## 1. Introduction

In the age of digital transformation, children are often exposed to psychological damage, abuse, and violence owing to a lack of internet monitoring. Even children who do not fall victim to the predatory behavior of adults often find themselves disadvantaged by the actions and omissions of services and products that do not consider their needs or take sufficient steps to protect them. With online selling and consumption becoming increasingly predominant in our daily lives, children and youth are also vulnerable consumers [1]. They lack control over their personal data and transparency in their use owing to children’s absence of the critical judgement required to identify circumstances of detriment in the digital environment. Such detriment includes cyberbullying, exposure to harmful content, the spread of mis- or disinformation (or “fake news”), and misleading or fraudulent commercial practices [2]. In particular, children may be exposed to potentially harmful marketing strategies, as well as being targeted with advertising based on their personal data, which raises privacy and financial or security concerns. With the advent of digital technologies such as artificial intelligence and predictive analytics, children’s data may also be used or abused for the purpose of profiling, potentially affecting their fundamental legal rights and freedoms. The increasing use of smartphones, digital assistants, and other mobile devices by children and youth raises issues of a lack of supervision by parents or guardians and a lack of adequate payment authentication and control tools. The Organization for Economic Cooperation and Development (OECD) has suggested recommendations on topics such as principles for a safe and beneficial digital environment for children, an overarching policy framework, international cooperation, and digital service providers [3]. 

Online safety is often used interchangeably with internet safety, cyber safety, internet security, online security, and cyber security. Risks related to online safety for children and youth include using computers, mobile phones, and other electronic devices to access the internet and social media, image-based abuse, cyberbullying, stalking, exposure to unreliable information or illicit materials, and breaches of privacy [2]. Although online safety is important for protecting children and youth from dangerous and inappropriate websites and materials, this does not mean that parents should discourage their children from using digital technology. In fact, the internet or any digital technology can transform children’s lives for the better, as it can open up a new world of entertainment and information and allow them to learn in new and unexpected ways [4]. The challenge is to help children and young people enjoy the benefits of going online while maintaining the skills and knowledge to identify and avoid risks. In this study, devices used by children and youth are classified as simple media and smart devices. Smart devices are those with online functions based on the added wireless internet. The online safety discussed in this study is not a problem with the device itself but rather a problem caused by the online connection through the device. This study first analyzes online safety research trends, followed by categorizing the types of online risks within the “4Cs framework”. Moreover, it conducts a case study on Australia, Canada, and the United Kingdom (UK), the three most advanced countries in terms of online safety-related policies for children and youth [5]. It attempts to focus on policy responses to “contract” within the “4Cs framework”. Finally, this paper presents policy recommendations that consider not only online risks in terms of the development of children and youth, but also the risks related to privacy and cybersecurity that they may experience while using digital technology or being involved in digital consumption. 

## 2. Literature Review

### 2.1. Perspectives on Children’s Use of Media or Smart Devices

Previous studies on media have focused on the negative effects of TV, video, and computer games on child development [6,7]. Television and videos that were watched extensively by children in the past are one-sided media without any interaction, which is different from smart devices that are frequently used in current times. Meanwhile, previous research on smart devices has shown that they promote creativity and autonomy in children and youth and have positive educational effects [8,9]. However, it has been identified that smart technology can pose a threat to the imaginative play of infants and children [10], and can negatively affect behavioral problems, academic performance, socialization, and language development [11]. The body of literature examining the risks associated with smartphone usage, including excessive use [12] and their impact on mental and physical health [13,14], as well as a loss of control [15], has been expanding. Furthermore, evidence suggests that children and adolescents are particularly susceptible to these risks. 

Currently, our children and youth belong to a generation that must live with smart devices, regardless of their pure function or dysfunction. According to a Korean study [16], the first time an infant used a smart device was early, with the proportions being highest at 45.1% for one-year-olds, 20.2% for two-year-olds, and 15.1% for three-year-olds. It was found that the timing of using smart media is getting faster [16], and it has become necessary to help children and youth use the device correctly rather than banning it unconditionally. Furthermore, reports from international organizations have shifted their focus from protection in smart devices to their appropriate use. Therefore, it is important for adults surrounding children to recognize the risks that children and youth may face in an online environment. 

### 2.2. Online Safety for Children and Youth

The concept of online safety has been relatively understudied in the realm of child development and early childhood education. While numerous studies have explored specific aspects of children’s and youth’s online behaviors, such as device usage patterns and the impact of digital content like games or social networking sites (SNS) on their development, there has been a limited focus on comprehensively examining the issue of online safety. Insufficient attention has been given to the research pertaining to the role of children and youth as consumers requiring protection within the online environment. Meanwhile, in the fields of business management and international trade, children and youth are discussed only as consumers. Since children and youth are the main consumers of digital media, they seem to be more interested in management and industry. However, since management and industry prefer to pursue profits, there is a weak point in protecting children and youth, especially children’s and youth’s privacy. We discuss online safety according to the best interests of children and their age-appropriate development. Reports from international organizations such as UNICEF, OECD, and APEC play a pioneering role in the online safety of children and youth. Relatively few online safety studies have been conducted on infants and children in academic fields; most studies concern children over eight years of age, with a preponderance on the youth [17]. In children and youth studies, considerable research is required on the risk of media devices; studies on the traditional risks before and after the online connection of digital devices are insufficient.

In 2012 and 2021, the OECD detailed the online risks that children may face (Figure 1). In the 2012 version, internet technology risks were subdivided into content, contact, consumer, and security risks. In the 2021 version, the category of risk was modified to include content, conduct, contact, consumer, and cross-cutting risks penetrating all categories. The OECD report has the advantage of considering children’s risks in terms of macroscopic aspects and others like consumers and technology. This study describes a category in which parents and teachers can easily consider online safety for optimal child development by referring to previous studies [2,18,19]. It assumes that the risks surrounding children and youth change significantly before and after the online connection of digital devices.

## 3. The 4Cs Risks of Online Safety for Children and Youth

As discussed earlier, since smart devices have been widely distributed for only 15 years, there has not been enough time to study their impact on children longitudinally and in depth. With a high internet and smartphone penetration rate, Korea had only 800,000 smartphone subscribers at the end of 2009, with the number exceeding 50 million in 2018; Korea’s population was about 51 million in 2018. Children and youth today are called digital natives, and their use of online media has changed rapidly from how they used it in the past. In a recent study, researchers proposed creating a new framework for a diverse audience regarding online risk [19]. Compared to prior research, our study places greater emphasis on content risk, particularly regarding infants and toddlers. We posit that the nature of the encountered content holds significant importance in ensuring online safety for this age group, surpassing its relevance for other age ranges. Figure 2 proposes the online risks faced by children and youth.

### 3.1. Contact Risk

Contact risk can be classified into contact risks for devices and harmful content. The younger the child, the more important the usage time and method of contact with the device; however, research on older children and youth does not seriously consider usage time. The smartphone is a highly passive device that requires simply sitting and becoming engrossed in information; it does not fit the early childhood development stage. According to a brain wave test, smartphone addiction among young toddlers reduces right brain function [20]. Excessive use and contact with a device can cause device addiction. And reducing the time spent on the device may reduce the time exposed to any online risk. Since the device is connected online, various unthinkable risks can emerge. Therefore, in this study, the risk of contact is divided into the device and content categories.

Recent digital devices that can be connected to the internet are mobile; therefore, they can be used regardless of the location. Parents sometimes use smart devices to neglect their children, even for infants and toddlers [21,22]; without appropriate restrictions, children and youth are likely to experience smart device addiction. Smart devices are acting as digital nannies. Videos that children and youth watch through the media has appealing elements like fast screen transitions, sounds, intense colors, and the appearance of animals and dolls [23]. Sites like YouTube have repeatedly used media devices for long periods of time by recommending videos suitable for viewers through YouTube AI and the YouTube Bot. Many children and youth come in contact with the device personally and can move it to their space at any time for usage, making it easy to deviate from parental guidance and supervision.

According to the displacement hypothesis, time spent on physical, social, and learning activities is reduced by the time spent using media [24]. Time spent using media is likely to replace children’s activities which are more appropriate for them and likely to stimulate their cognitive development [25,26]. The longer children watch TV, the less time they spend on creative games like drawing, playing musical instruments, pretend play, and toy games [27]. A study of Korean children found that mothers’ excessive use of smart devices as a means of neglecting their children lowers the children’s language development [21,22]. In a recent study, the screen time of primary school students is the main predictors of children’s cyberbullying and online grooming involvement, together with parental supervision [28].

### 3.2. Content Risk

Along with mobility—a characteristic of smart devices—non-sharing weakens adult supervision of smart devices. Children and youth generally use smart devices alone rather than with others; therefore, they are likely to be exposed to harmful content. Ultimately, children and youth became more easily exposed to harmful content that is not suitable for development than before having online connections. Content that is not suitable for development is raunchy or violent, which negatively affects children and youth. A developmental perspective is essential for an adequate understanding of how media violence affects youthful conduct, and for formulating a coherent response to this problem [29]. 

During childhood, children encode social scripts in their memory to guide their behavior through the observation of family, peers, community, and mass media. Consequently, these observed behaviors are imitated long after they are noticed [30]. This is the main idea of a social cognitive model; repeated exposure to violent or cruel scenes makes negative emotions habituated, and the child becomes “desensitized”. Subsequently, children can consider and plan proactive aggressive acts without experiencing negative effects [31]. Pornography exposure has been correlated with nonrelational or recreational sexual attitudes and behaviors in previous studies [32]. According to previous studies, parental indirect intervention can alleviate the relationship between the degree of violent video exposure and the aggression of children [33]. Adults surrounding children and youth need to manage and check the content they consume through smart devices for their safety.

### 3.3. Conduct Risk

Social networking services (SNSs) are positive means of intimacy and expression when used appropriately by people. However, if used incorrectly, these can be used as a means of attacking and criticizing someone indefinitely by hiding behind the convenience of non-face-to-face and anonymity, and can also put others at risk [34]. Many risks faced by children and youth are online versions of well-known offline risks (bullying, racism, cheating, and sexual predation) [35]. However, parents and teachers should understand that online versions are more threatening than offline ones, because they are not limited to time, space, or target. In online spaces, victims can be bullied anonymously, and if they have access to online spaces, such harassment can occur indefinitely without space-time constraints or adult control [36,37]. This characteristic of an online space is referred to as an unlimited large audience without restrictions on the number and role of participants.

In the online space, more open self-expression is possible by releasing self-control devices that have been suppressed in reality, leading to guiltless and harsh acts of violence [38,39], called the online disinhibition effect. Online connections with countless disinhibited individuals increase the probability of children becoming victims or perpetrators of crime. It is easier to anonymously attack someone online than offline, and distinguishing them becomes meaningless as they are both perpetrators and victims. 

Children and youth are more likely to contact harmful peers or adults in online circumstances than in offline ones. In the offline version, the environments in which children and youth meet bullies or criminals are relatively limited. However, it is no exaggeration to state that the risk of the online version continues in infinite space and is perpetrated indefinitely. Previous reports on this topic have usually focused on sexual problems, but there are only a few academic papers on this topic. This is a clear crime; however, it has not been long since these crimes appeared. Therefore, laws, social systems, and research have not been able to keep pace with the crime. According to the poly-victimization theory [40], victims are followed by other forms of abuse rather than by a single episode. In other words, if a child is the target of one crime, they are more likely to suffer other types of abuse. Consequently, these risks cause various problems in children and youth, such as internalizing and externalizing problems, suicidal ideation, and a poor quality of life [41,42]. Such negative contact may also occur in ways like contacting someone for goods transactions or part-time jobs, causing financial and emotional damage, or being involved in a fraud.

### 3.4. Contract Risk

Children are not equipped with perfect cognitive skills compared to adults during the developmental process, which demands protection by their parents and teachers. In the long run, children grow into consumers who influence future economic activities. Although they use the internet from childhood, there are only a few studies involving young children in establishing the appropriate levels of regulation or educational policy for their development [17]. This suggests that not only young children but also parents and teachers are least bothered about this issue. Children and youth have rights that must be protected from violation and they should be able to enjoy the internet, free from manipulative and exploitative practices [43]. Privacy is an abstract and complicated concept, whose norms are in flux, making it difficult to impart clear, relevant, consensus-based messages [44]. The study of media literacy in children related to their data rights and privacy is rarely found.

Children are an extraordinarily powerful consumer group equipped with the technology to exercise commercial influence while also wielding a persuasive influence over their parents’ buying choices [43]. Online services that use children’s personal information can turn them into commercial targets, which is likely to adversely affect children and youth by encouraging reckless consumption. Moreover, unlike parents and adults, children and youth have been exposed to online devices from a very young age, before experiencing any social interaction. Since it has not been confirmed how they will grow and develop in the future, policies targeting children and youth should be conservative and protective. To deal with the threats created by “sharing”, a child’s privacy must be considered not only as a right but also in the “child’s best interest” [45]. Specifically, in collecting the personal information of children and youth, the purpose of collecting and using the data should be stated, items should be accurately notified, minimum information should be collected, and personal information should not be used or transferred for other purposes. In addition, simple and understandable forms and languages should be used, and more protective and careful measures are needed when targeting adults, such as confirming the consent of legal representatives. Children and youth are particularly vulnerable to targeted “stealth and social marketing”, hence needing additional special protection from digital manipulation technologies that are stronger than those for adults [46]. Children must not be treated as simply another consumer group to be exploited or avoided by the industry. It is time to formalize and strengthen constraints on advertising to ensure that consumers’ best interests precede innovation and monetization [43].

## 4. Case Study—Australia, Canada, and the UK

### 4.1. Australia

#### 4.1.1. Governance and Legislation

In Australia, the Office of the eSafety Commissioner (the Office hereafter) is an independent statutory office created by the Enhancing Online Safety for Children’s Safety Act 2015, and it is meant to coordinate and lead online safety efforts across the government, industry, and civil society. In addition, the Criminal Code Act 1995 captures a wide range of offences relevant to the online environment for children and youth. Although the Office is not child-specific, it has a strong focus on children and is responsible for the general oversight of children in the digital environment, administering complaint schemes, accrediting/training educators, directing the removal of online content, and issuing sanctions [5]. The Office places online harm into four categories: illegal and restricted online content, image-based abuse, cyberbullying, and adult cyber abuse. The former two are covered in content risk, and the latter two come under conduct risk within the 4Cs risk framework. The Office operates the eSafety Hotline for parents and caregivers who encounter illegal content online, and tackles image-based abuse through an online portal and reporting tool. To support schools, teachers, parents, and caregivers in protecting children, the Office further provides expert knowledge applicable to parents and educators worldwide, including a range of practical tips for topics including media, misinformation, scams, time spent online, parental controls, unwanted contact, cyberbullying, online gaming, and advice on self-care. Moreover, the Office provides access to an online safety booklet for children under five which also includes advice on parental controls to set up filters on the home internet. This includes practical and specific messages like applying parental controls to limit screen time and block specific app use and websites, instructions on how to limit use of the camera and microphone to prevent external communication, applying age restrictions to media content and websites, monitoring a child’s use of apps or web browsing activities, configuring web browsers to use “safe search”, and ensuring that children use devices in sight of their parents. 

To respond to risks related to privacy and cybersecurity, the Office of the Australian Information Commissioner (OAIC) was established as an independent national regulator for privacy and freedom of information, promoting and upholding rights to access governmental and personal information [5]. It is responsible for disseminating knowledge of the government’s guidelines and principles regarding privacy awareness, having regulatory powers to investigate complaints, and to enforce compliance with Australia’s privacy principles. Digital providers conducting businesses in Australia, like Facebook, Twitter, Instagram, and YouTube, should specify under law that users must be at least 13 years old, although Australian parents are usually unaware of this requirement. By specifying minimum age stipulations, many social networking sites should gain verifiable parental consent prior to collecting any personal information from a child younger than 13 years, providing their services without any penalty for disobedience. Consent to share personal information is required for children aged < 15 years [5]. Under the National Office for Child Safety, the government further developed the National Principles for Child Safe Organizations to recognize the importance of safe physical and online environments to promote the well-being of all children and young people. The Australian Cybersecurity Center (www.staysmartonline.gov.au) provides information on cybersecurity for Australian internet users without a specific space for children and youth, while the Department of Education provides online safety education to children, parents, guardians, and teachers, operating a “Safe Schools Hub”, a one-stop shop for relevant information and resources. In addition, the Australian Communications and Media Authority (ACMA) provides information on internet safety to parents and caregivers.

#### 4.1.2. Major Programs

The office provides “eSafety information”, a hub of information about e-safety issues and how to protect users, including children and youth, and their personal information. “Stay Smart Online”, a one-stop shop for Australian internet users, provides information on the simple steps to protect their personal and financial information online, including informative videos, quizzes, and a free-alert service that provides information on the latest threats and vulnerabilities. “The School A to Z website” by the New South Wales Department of Education (or the NSW government) provides practical help to parents regarding keeping kids safe online, including tips for keeping one’s family’s personal information safe. “Schools and Cybersafety” by the Victorian Department of Education and Training advises schools on cybersecurity and the responsible use of digital technologies. It encompasses a range of topics including bullying, cybersecurity strategies, and practical steps and actions related to online incidents. Among the risks associated with contact risk, issues related to gambling are key. Accordingly, the Australian Institute of Family Studies (AIFS) has introduced “Gambling Help Online”, a short-term online service developed to support family and friends impacted by this problem.

### 4.2. Canada

#### 4.2.1. Governance and Legislation

The Canadian Center for Child Protection (C3P) deals with the issue of online safety, offering information about the ever-changing online interests of children, the potential risks they face, and proactive strategies to protect them online. C3P is a national charity dedicated to the personal safety of children with the goal of reducing sexual abuse and exploitation, assisting in locating missing children, and preventing child victimization. The C3P operates “Cybertip.ca (accessed on 15 June 2023)”, a national tipline for reporting child sexual abuse and exploitation on the internet, as well as other intervention, prevention, and education services to the Canadian public. This website helps Canadian students with concerns about shared intimate images, online luring, and other areas involving child victimization. The government also organizes “Safer Internet Day” (SID) every February, where people and organizations globally join forces for SID to promote the safer and more responsible use of online technology and mobile phones, especially among children and youth, with C3P playing the role of coordinating SID information and activities. Related to privacy and cybersecurity, Canada requires that consent for sharing personal information be fully informed, implying a need to use child-friendly and appropriate language in seeking consent, and an implicit ban on obtaining consent from children of a very young age [5]. Created in 2003 to ensure coordination across all federal departments and agencies responsible for national security and the safety of Canadian citizens, Public Safety Canada plays a significant role in seeking input from individuals and organizations to contribute knowledge and ideas related to the cybersecurity and cybercrime landscape. It has three mandates: the National Cyber Security Strategy to protect citizens and organizations from cyber threats, the National Cyber Security Action Plan as a blueprint for the implementation of the Strategy, and the Cyber Security Cooperation Program to support projects through grants and contributions to improve the security of Canada’s vital cyber systems.

#### 4.2.2. Major Programs

The online programs provided by C3P focus on parents, children, and youth. For parents, C3P provides “ProtectKidsOnline.ca (accessed on 15 June 2023)”, a website which regularly gleans information from Cybertip.ca to help parents stay informed about the age-specific interests of young people, the risks they face online, and proactive strategies to make their child’s online experiences safer, while “Self/Peer Exploitation” helps parents and educators respond to incidents of self/peer exploitation, better known as sexting. For children, “Zoe and Molly Online”, an interactive website series, provides an opportunity for 8–10-year-olds to have some fun exploring what it means to be safe while playing games online. Using comics, interactive games, and online safety quizzes, the website provides children with an engaging learning experience. For the youth, “NeedHelpNow.ca (accessed on 15 June 2023)” provides information about managing issues that may arise from sexting, as well as steps to take to request that images/videos be removed from websites, helpful tips on involving a safe adult, information about self-care, and recognizing when things have gone too far. Furthermore, “DontGetSextorted.ca (accessed on 15 June 2023)” not only educates tweens and teens about sextortion but also provides a unique way to prevent it by providing downloadable naked mole rat gifs and memes—the perfect alternative to send when asked for a nude. “Project Arachnid” is an automated web crawler and platform to help reduce the online availability of child sexual abuse material worldwide, while “NeedHelpNow.ca” helps teens to stop the spread of sexual pictures or videos and provides support along the way, along with complementing resources. Finally, “DontGetSextorted.ca” is a website which tackles the issue of sextortion and how teens can prevent this from happening to them.

### 4.3. UK

#### 4.3.1. Governance and Legislation

The UK Council for Internet Safety (UKCIS), guided by the government’s Internet Safety Strategy, is the key governmental body working to improve online safety, particularly that of vulnerable groups like children and youth, who are often disproportionately targeted for online abuse. Until November 2018, the UK Council for Child Internet Safety (UKCCIS) focused solely on children’s issues and played a pioneering role in promoting and championing online safety for children and youth. Its mandate was outlined in the Government’s Internet Safety Strategy Green Paper in October 2017. The UKCIS, the new council, has specific objectives reflecting children and young people’s special needs for care and protection, building on the pioneering work of the UKCCIS in this area, as the Executive Board (the Board hereafter) contains representatives of children’s organizations. The Board consists of three co-ministerial chairs from the Department for Digital, Culture, Media, and Sport, the Home Office, and the Department for Education. The UKCIS is a multi-stakeholder forum with an interest in online safety and partnerships with 200 organizations representing the government, regulators, industry, law enforcement, academia, and charities [5]. Bringing together key stakeholders and working across sectors and disciplines is supposed to build a safer internet that can integrate the experiences of a wide variety of citizens [4]. The UKCIS particularly focus on online safety risks such as cyberbullying and sexual exploitation, radicalization and extremism, violence against women and girls, hate crime and hate speech, and forms of discrimination against groups protected under the Equality Act, for example based on disability or race. 

To respond to risks related to privacy and cybersecurity, the UK’s 2018 Data Protection Act includes a provision requiring the introduction of an “Age-Appropriate Design Code” (the Code hereafter). Launched in April 2019, the Code provides requirements that online services must meet to make their services available for children and youth, with compliance monitored by the Information Commissioners Office (ICO). As the ICO is an executive non-departmental public body sponsored by the Department for Digital, Culture, Media, and Sport (DCMS), it upholds information rights in the public interest, promoting openness by public bodies and data privacy for individuals. Unfortunately, the ICO has the power to issue warnings, reprimands, stop-now orders, and fines for breaches in the General Data Protection Regulation (GDPR). The provisions in the draft code only require online services to provide an age-appropriate service and be transparent to allow children to understand the information presented to them, uphold community standards and the providers’ published terms of service, ensure that geolocation tracking is off by default for children, and ensure the safety, security, and privacy of children where smart or connected devices are used at home [5]. To respond to risks related to privacy and cybersecurity, the DCMS provides a guideline on child online safety for data protection and privacy that is intended for organizations that provide online services likely to be accessed by children.

#### 4.3.2. Major Programs

The UKCIS provides “Wi-Fi logos” which enable the identification of public Wi-Fi spots that have filtered inappropriate websites. This website also collates internet safety research and creates guides for parents with practical safety and privacy tips, and for the industry, which include examples of good practices and advice from online child safety experts. The government launched the Verification of Children Online (VoCO) project, a child safety research project, in collaboration with the DCMS, Home Office, and Government Communications Headquarters (GCHQ). The project has joined children, industry, and child safety stakeholders to consider the technical, commercial, legal, and behavioral factors that would enable companies to recognize and better protect their child users. Table 1 is the summary on online safety policies in three countries.

## 5. Analysis and Policy Implications

This study was conducted to identify the risks faced by children and youth, and to establish online safety at a time when their use of online smart devices is increasing daily. Previous studies on children and youth have focused on examining the effects of smart devices on children’s development or interventions at the family level. Additionally, microsystem-level research has been conducted, such as education on online safety in educational institutions. Therefore, few studies have focused on the macro perspective of the state, law, policy, cases of other countries, and international cooperation. Based on the 4Cs framework used in existing studies, this study examined the online risks faced by children and youth. Considering this, we revised the 4Cs risk framework and compared cases from advanced countries related to online safety. Thus, we attempted to obtain the implications for online safety-related policies (Figure 3).

Two key policy implications were derived from the comparative policy analysis of the target countries: the multi-stakeholder and multilevel policymaking approaches. First, the target countries adopted a multi-stakeholder approach to enhance online safety for children and youth, which required a joint approach from various stakeholders (Table 2). The major stakeholders were parents and educators (teachers/professionals), schools and private installations, policy-makers’ integration and regulation, digital finance as well as ICT industries, and children and youth who were interested. Online safety issues related to digital device use, such as media addiction, should be addressed from the perspective of the entire domestic and global system surrounding children and youth rather than as an individual’s or family’s problem. It was suggested that what children and youth do online has more bearing on their well-being than how much time they spend online; children who are more active online are also better at managing online risks [4]. Therefore, it is better for adults to design and implement policies to effectively facilitate online experiences, rather than hinder children’s internet use. There should be a systemic approach that involves not only the domestic policy framework of each country, but also an international policy framework that includes states and relevant IOs to design and establish laws and policies, conduct international cooperative projects, and establish industry guidelines for digital firms. Based on this, each government needs to request appropriate policy cooperation from domestic or multinational digital firms for the true well-being of children and youth.

Second, a multilevel policymaking approach is required that includes three layers: national legal and policymaking, which comprises legislative responses and policy instruments (direct and indirect); multi-stakeholder policymaking, which is related to the different roles and responsibilities of stakeholders; and international policymaking, which comprises cross-operation and initiatives targeting knowledge sharing [4]. In this regard, the role of policymakers is to monitor the structure of administration and regulation to enhance online safety for children and youth. It is desirable for each country’s policy promotion entity to be in the form of a one-stop administrative agency, like in major advanced countries. For example, Australia has established the Office of the e-Safety Commissioner to oversee children and youth in the digital environment, administer complaints, provide schemes, accredit, or train educators and professionals. Canada has the Canadian Center for Child Protection (C3P), while the UK has the New Council for Internet Safety since 2008. In addition, the age for online consumption participation is harmonized in all three target countries at 13, but the working age under parental permission, which secures more income and digital consumption, differs across countries or states at 13 or 15. International policy coordination on these issues will enable more consistent policy implementation for the better online protection of children and youth.

Moreover, due to the nature of digital products, service distribution, and trade, measures to improve online safety for children and youth are limited domestically. International joint efforts are essential for “contraction” related to privacy and cybersecurity through the cross-border transfer of data. Currently, in bilateral negotiations in the Trade and Technology Council (TTC) between the US and the EU, Working Group 6 taps into the issue of the “misuse of technology threatening security and human rights”, which attempts to counter cyber threats and technology used to violate human rights, as well as to address those conducting information/disinformation operations. From the industrial policy perspective of each country, the responsibilities and obligations of a digital service provider related to the online safety of children and youth are increasingly emphasized, as these consumers are recognized as important participants in the digital service industry and consumer protection (Figure 4). Although digital platform companies may be separate from digital content development, there is criticism that they account for a high proportion of total sales, and control individual digital transactions; occasionally delivering distorted information such as fake products and false reviews without filtering. It is necessary to both improve awareness within a country and cooperate internationally, as children and youth are increasingly affected by the global digital market and policy environment.

Each country must prepare more carefully for domestic systems related to the privacy and cybersecurity of children and youth consumers at each stage of economic and digital trade development [47]. There are two main approaches to protecting children’s privacy in an online environment: protecting their vulnerability to online risks by empowering them to enhance their capability of recognizing online risks and exercising their own privacy-related decisions. Relevant legislation in the US, the EU, and Korea focuses on protecting children through parental guardianship. This approach, with the consent of parents or schools, could result in profoundly unequal restrictions on children’s and youths’ active rights to access the internet freely. Moreover, if children aged 12–13 or older and 16–17 years old are considered children and youth, the seriousness of such infringements can be evaluated further. Thus, it is necessary to clarify the development of “data literacy” for securing online safety of children and youth, enabling them to properly exercise their right to personal information protection, and actively seek ways of empowering the children and youth. Likewise, it is recommended to refer to China’s regulatory trend of controlling issues related to the privacy infringement of minors under its recent legislation on personal information and data protection. 

Finally, it is suggested that families pay attention to the digital device usage times of their children and youth. Parents, especially those in families with infants and toddlers, should first limit their own device usage time, as the adverse effects of digital devices are obvious. Parents or teachers should be interested in online platforms or apps used by children and youth and suggest discussing and sharing online activities together. This is an effective way to use the platforms and apps that their children use. Indeed, a recent study argued that teenagers positively recognize parents’ active mediation, monitoring, and participation in online safety in respect to family communication and suggest that teens value parental involvement and do not desire complete independence online [48]. Continuous communication and consultation with children is imperative in the context of online safety. At the school level, it is crucial to present and educate students on digital devices, privacy, personal information, and the digital risks of the 4Cs. Australia’s National Framework Study for Online Safety Education proposed an effective whole-school approach suggesting that online safety education is underpinned by effective whole-school approaches for promoting student wellbeing and preventing student harm [49]. 

## 6. Conclusions

To minimize the four major online risks for children and youth in cyberspace, by conducting a comparative study of Australia, Canada, and the UK, this paper argues that it is necessary to maintain a regulatory approach to the online exposure of children under the age of 13. Moreover, this paper suggests that the global society should respond together to these online risks with “multi-level” policymaking under a “multi-stakeholder approach”. At the international level, the OECD and those under UN subsidiaries should continue to lead global discussion and promote international cooperation. At the domestic level, a special agency in charge of online safety for children and youth should be established in each country, encompassing all relevant stakeholders and bridging educators and digital firms. At the school and family levels, both teachers and parents need to work together in facilitating digital literacy education and providing proper guidelines for the online activities of children and youth. Prior research highlights the necessity of implementing technical interventions that effectively enable appropriate control measures while maintaining trust with children [50]. In practical settings, some studies have proposed a privacy-preserving parental control protocol with edge computing that uses artificial intelligence techniques to automatically detect harmful content for minors in 5G networks [51]; in another example, a new parent–children blockchain-based supply chain data supervision system is proposed, which aims to overcome the dilemma faced by the governmental regulation of supply chains [52]. Child protection apps need to consider functions such as automatically removing restrictions and monitoring as the children grow up. According to a previous study analyzing parental control apps for online safety promotion, child protection apps for parental intervention have an essential problem of focusing only on effective restrictions and monitoring instead of on digital parenting [53]. To enhance online safety for children and youth, discussion is needed not only within the education field but also the fields of business and economics, as well as that of engineering and technology. Academics must collaborate on elaborating digital risks and conduct longitudinal studies on the effects of device use in children and youth. Since the advent of digital devices was less than 20 years ago, their influence on the development of children and youth should be carefully assessed. Studies thus far have shown that the parental use of smartphones may be related to negative changes in parental sensitivity and responsiveness, and these low levels of parental sensitivity and responsiveness have an adverse effect on their children [54]. Adults surrounding children should not forget that smartphone and tablet use is related to poorer overall early childhood factors [55]. As laws and systems to prevent crimes related to the 4Cs have not been overhauled, measures should be implemented and international cooperation should be strengthened immediately. The use of smart devices by children and youth is not a problem that individual families or parents can control. Therefore, these measures are necessary not only for protecting individual children and youth but also the society that future generations will create. 

## Figures and Tables

**Figure 1 children-10-01415-f001:**
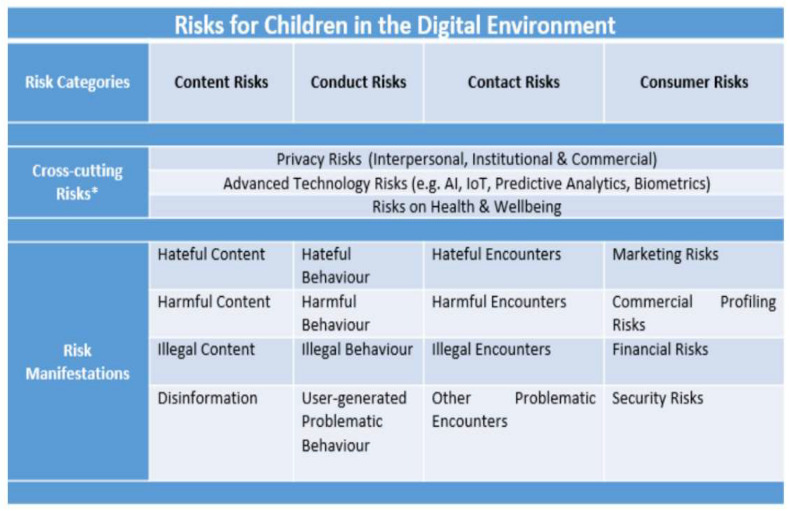
Typology of risks (OECD, 2021). * Note: The Typology acknowledges risks that cut across all risk categories (“Cross-cutting risks”). These risks are considered highly problematic as they may significantly affect children’s lives in multiple ways.

**Figure 2 children-10-01415-f002:**
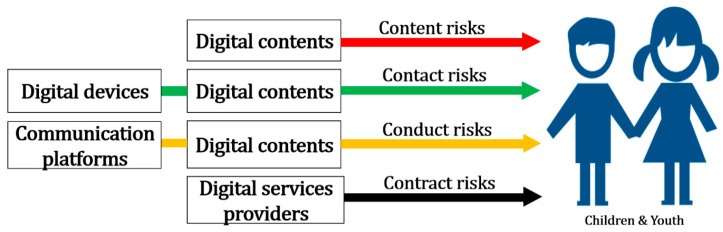
The 4Cs risks of online safety for children and youth.

**Figure 3 children-10-01415-f003:**
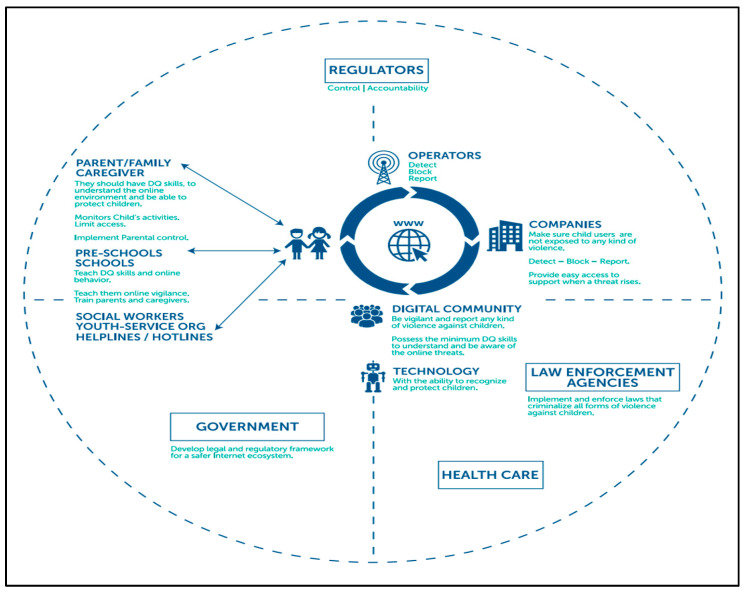
Domestic environment of online safety for children and youth source: UNESCO (2019).

**Figure 4 children-10-01415-f004:**
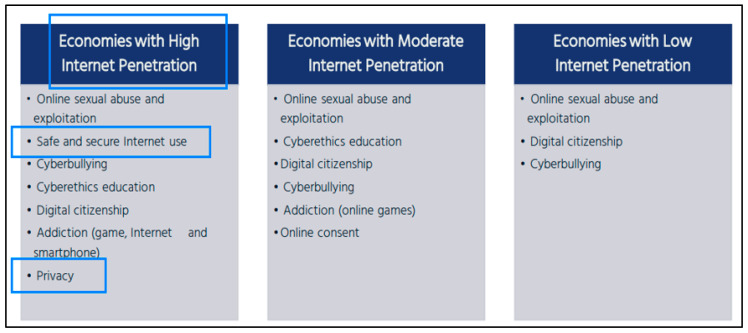
List of data protection policy issues within economies source: Internet Society (2017).

**Table 1 children-10-01415-t001:** Summary on online safety policies for children and youth in three countries.

Country <Basic Information>	National Legislation	Governance and Government-Led Programs
Australia <total population: 25,739,256 (2021) internet users, of population: 90% (2020)>	The Enhancing Online Safety Act (2015) The government established a range of measures to improve the online safety of children in 2014 including a public consultation in January 2014	The Office of the e-Safety Commissioner—General oversight of children in the digital environment, administers complaints scheme, accredits/trains educators, can direct the removal of online content and issue sanctions (Not child-specific, but with a strong focus on children) The Office of the Australian Information Commissioner (OAIC) to respond to risks related to privacy and cybersecurity Australian Cybersecurity Center (www.staysmartonline.gov.au (accessed on 1 May 2023)) provides information on cybersecurity for Australian internet users and has a specific space for the youth The Department of Education makes online safety education available to children, parents, guardians, and teachers, and operates the “Safe Schools Hub”, a one-stop shop for relevant information and resources The Department of Communications The Australian Communications and Media Authority (ACMA) provides information on internet safety for parents and carers
Canada <total population: 38,246,108 (2021) internet users, of population: 97% (2020)>	The Consumer Privacy Protection Act The Personal Information and Data Protection Tribunal Act	The Canadian Center for Child Protection (C3P) Public Safety Canada on cybersecurity The Office of the Privacy Commissioner of Canada (OPC) Canadian Centre for Cyber Security Innovation and Skills Plan Multiple Departments including Innovation, Science, and Economic Development
UK <total population: 67,326,569 (2021) internet users, of population: 95% (2020)>	Online Safety Bill (2019)	New Council for Internet Safety in the UK (from 2008) Department for Digital, Culture, Media and Sport (DCMS) Government Communications Headquarters (GCHQ)

**Table 2 children-10-01415-t002:** Major issues and domestic stakeholders of online safety for children and youth.

Major Actors or Stakeholders	Enabling Policies on Protection of Children Online	International Cooperation on Protection of Children Online	Managing Business Practices	Improving ICT Access	Improving Digital Competencies	Targeted Education and Awareness Campaigns
Policymakers	O (National policy)	O	O	O	O	O
Digital firms	O (Industrial policy)	O	O	O	n/a	n/a
Educators	O (Education policy)	O	n/a	n/a	O	O
Parents/carers	O (Family policy)	n/a	n/a	O	O	n/a
Children and Youth	O (Education policy)	n/a	n/a	n/a	O	O

n/a: not applicable.

## Data Availability

Not applicable.

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
