# Peer review of "Online Safety for Children and Youth under the 4Cs Framework—A Focus on Digital Policies in Australia, Canada, and the UK"

_children, 2023, doi:10.3390/children10081415_

Round 1
Reviewer 1 Report
The article has a positive point of view although it is talking about risks. The authors have tried to balance different ideas about the use of devices.
The title should be more appropriate. The article is comparing different country laws. It is not clear if they are talking about consumers related to advertising or devices consumers.
Objectives must be clarified to understand the article.Some of the sentences, in conclusions have not confirmed. It looks like opinions.
I suggest to clarify the objectives of the research and try to join them with conclusions. The article is not clear about the research goal.
Author Response
[Reviewer 1]
1)The article has a positive point of view although it is talking about risks. The authors have tried to balance different ideas about the use of devices.
=> We appreciate for your thorough review and positive opinion of our manuscript.
2) The title should be more appropriate. The article is comparing different country laws. It is not clear if they are talking about consumers related to advertising or devices consumers.
=> => Editing We decide to change the title as “Online Safety for Children and Youth under the 4Cs Framework - A Focus on Digital Policies in Australia, Canada, and the UK”
3) Objectives must be clarified to understand the article. Some of the sentences, in conclusions have not confirmed. It looks like opinions.
=> Adding Information We tried to supplement the conclusion.
Adults surrounding children should not forget that Smartphone and tablet use is related to poorer overall early childhood factors [54].
4) I suggest to clarify the objectives of the research and try to join them with conclusions. The article is not clear about the research goal.
=> Adding Information We decided to add more concrete summary of our paper in the introductory part of the conclusion.
To minimize the four major online risks for children and youth in cyberspace, by conducting a comparative study of Australia, Canada, and the UK, this paper argues that it is necessary to maintain a regulatory approach to the online exposure of children under the age of 13. Moreover, this paper suggests that the global society should respond together to these online risks with “mul-ti-level” policymaking under a “multi-stakeholder approach." At international level, the OECD and under UN subsidiaries should continue to lead global discussion and promote international cooperation. At domestic level, a special agency in charge of online safety for children and youth should be established in each country, encompassing all relevant stakeholders and bridging educators and digital firms. At school and family levels, both teachers and parents need to work together facilitating digital literacy education and providing proper guidelines for online activities of children and youth.
Studies so far have shown that parental use of smartphones may be related to negative changes in parental sensitivity and responsiveness, and the low levels of parental sensitivity and responsiveness have an adverse effect on their children [55].

Reviewer 2 Report
The theme of the study is quite relevant, taking into account the times in which we live.
There are some ideas in the text that need authors to support the ideas. For example:
“however, research on older children and youth does not seriously consider usage 151 time. Excessive use and contact with a device can cause device addiction, and reducing 152 the time spent on the device may reduce the time exposed to any online risk. Since the 153 device is connected online, various unthinkable risks can emerge. Therefore, in this study, 154 the risk of contact is divided into the device and content categories.”
or
“Parents sometimes use smart devices to neglect their children; without appropriate restrictions, children and youth are likely to experience smart device addiction.”
Comparing the digital policies of the 3 countries is quite relevant.
The references used may be updated, as this theme is constantly being updated.
Author Response
[Reviewer 2]
1) The theme of the study is quite relevant, taking into account the times in which we live.
=> We appreciate for your thorough review and positive opinion of our manuscript.
2) There are some ideas in the text that need authors to support the ideas. For example:
“however, research on older children and youth does not seriously consider usage time. Excessive use and contact with a device can cause device addiction, and reducing the time spent on the device may reduce the time exposed to any online risk. Since the device is connected online, various unthinkable risks can emerge. Therefore, in this study, the risk of contact is divided into the device and content categories.”
Or “Parents sometimes use smart devices to neglect their children; without appropriate restrictions, children and youth are likely to experience smart device addiction.”
=> Editing Smartphone is a highly passive device, it does not fit the early childhood development stage, that requires simply sitting and engrossing information. According to a brain wave test, smartphone addiction among young toddlers reduces right brain function [21].
=> Adding Information As you noted, this is one of our opinions. We think that online connectivity has become a very big risk for children and youth. With the online connectivity, children and youth have encountered more harmful content (violent and sexual photos and videos) that are not age-appropriate, and the scope is very wide. In the past, when there was no online connection, there was little risk to the content if the content inside the computer or smart device was blocked well. But now children and youth are exposed to dangerous content on servers around the world without filtering.
=> Adding Information Related previous studies have been supplemented.
Parents sometimes use smart devices to neglect their children, even for infants and toddlers [00-00]; without appropriate restrictions, children and youth are likely to experience smart device addiction. Smart devices are acting as digital nannies.
3) Comparing the digital policies of the 3 countries is quite relevant.
=> We appreciate for your thorough review and positive opinion of our manuscript.
4) The references used may be updated, as this theme is constantly being updated.
=>Adding Information We tried to refer to the latest studies related to this field. Previous studies were reviewed again, and interpretations and reports were revised.
Tintori A; Ciancimino G; Bombelli I; De Rocchi D; Cerbara L. Children’s Online Safety: Predictive Factors of Cyberbullying and Online Grooming Involvement. Societies, 2023, 13(2), 47. https://doi.org/10.3390/soc13020047
Walsh, K.; Pink, E.; Ayling, N.; Sondergeld, A.; Dallaston, E.; Tournas, P.; ... & Rogic, N. Best practice framework for online safety education: results from a rapid review of the international literature, expert review, and stakeholder consultation. International Journal of Child-Computer Interaction, 2022, 100474.
Seetharaman, R.; Rajeswari, PS. Smartphone Usage and the Addiction Behavior among Children-A Global Study. Special Education, 2022, 1(43), 6533-6541.
Mallawaarachchi, S.; Anglim, J.; Hooley, M.; Horwood, S. Associations of smartphone and tablet use in early childhood with psychosocial, cognitive and sleep factors: a systematic review and meta-analysis. Early Childhood Research Quarterly, 2022, 60, 13-33.
Braune‐Krickau, K.; Schneebeli, L.; Pehlke‐Milde, J.; Gemperle, M.; Koch, R.; von Wyl, A. Smartphones in the nursery: Parental smartphone use and parental sensitivity and responsiveness within parent–child interaction in early childhood (0–5 years): A scoping review. Infant Mental Health Journal, 2021, 42.2: 161-175.

Round 2
Reviewer 1 Report
The comments have been solved.